# Intelligent Discrete Sliding Mode Predictive Fault-Tolerant Control Method for Multi-Delay Quad-Rotor UAV System Based on DIECOA

Pu Yang [1,*], Zhiqing Zhang [1] , Huiling Geng [1] , Bin Jiang [2] and Xukai Hu [1]

[1] Department of Automation, Nanjing University of Aeronautics and Astronautics, Nanjing 210016, China; zhiqingzhang@nuaa.edu.cn (Z.Z.); hlgeng@nuaa.edu.cn (H.G.); xkhu@nuaa.edu.cn (X.H.)
[2] The National Key Laboratory of Science and Technology on Helicopter Transmission, Nanjing University of Aeronautics and Astronautics, Nanjing 210016, China; binjiang@nuaa.edu.cn
* Correspondence: ppyang@nuaa.edu.cn

**Abstract:** This paper introduces a novel intelligent sliding mode predictive fault-tolerant control method based on the Dynamic Information Exchange Coyote Optimization Algorithm (DIECOA), which is applied to a quad-rotor UAV system with multi-delay and sensor fault. First, the system nonlinearity and sensor fault are dealt with by means of interpolation transformation and system state expansion, and an equivalent system is obtained. Second, the quasi-integral sliding mode surface is used to construct the prediction model so that the initial state of the system is located on the sliding mode surface, and the global robustness is guaranteed. Third, this paper introduces an improved fault and disturbance compensation term, which effectively weakens the adverse effect of time delays and enhances the FTC performance of the system. Fourth, the Dynamic Information Exchange (DIE) strategy is designed to further improve the coyote individual replacement mechanism and speeds up the solution and convergence speed of the method in this paper. Finally, the simulation is carried out on the fault-tolerant simulation platform of the quad-rotor Unmanned Aerial Vehicle (UAV), and the results show the rationality of the method.

**Keywords:** fault-tolerant; sliding mode prediction; multi-delay; UAV; DIECOA

## 1. Introduction

In recent years, the rapid development of control theory and industrial automation has made the requirements for high-performance control methods more and more stringent [1,2]. However, traditional control methods have gradually failed to meet the needs of high-tech control systems, especially in aerospace control systems, such as quad-rotor UAV (Unmanned Aerial Vehicle) systems. Currently, quad-rotor UAVs are widely used in various fields due to their lightweight, flexibility, and ease of operation. However, the ubiquitous parameter uncertainties and external disturbance in the quad-rotor UAV system will make the initial environment of the system very complicated. In addition, incalculable losses will occur in the quad-rotor UAV system when specific components, such as actuators and sensors, inevitably break down due to long-term work [3,4]. In conclusion, how to design an FTC (fault-tolerant control) method that can solve the above problems in the quad-rotor UAV system is crucial [5,6].

There have been a large number of studies at home and abroad on robust fault-tolerant control. FTC is the ability of a system to maintain its original performance and continue to operate after a fault occurs, possibly at a reduced level of performance depending on the severity of the fault. Sensor fault generally affects the control effect indirectly, and it is usually the wrong measurement value that causes the system to produce the wrong control output. The actuator fault generally directly affects the control effect. The additive fault of the actuator refers to the deviation of the output torque of the actuator from the normal

value, and the multiplicative fault of the actuator affects the output gain of the actuator. The FTC system refers to a kind of control system that uses a new control method to deal with the existing faults of the system while maintaining the robust stability and ideal state of the system. Researchers have proposed many techniques, the traditional methods include PID, $H\infty$ control [7,8], and the emerging methods include predictive control, sliding mode control(SMC), adaptive control, and the fuzzy neural network [9–12].

In the above methods [9–12], the combination of different methods will also produce nice FTC performance, such as adaptive sliding mode and sliding mode prediction (SMP) [13,14]. Ref. [15] proposes an adaptive SMC allocation scheme that can maintain the tracking performance of the system by adjusting the control gain of the high-order module in the case of the same group of actuators failing simultaneously. The rationality of the method is verified on the octocopter. However, this article has not considered time delays. Ref. [16] introduces an SMC method based upon an adaptive fuzzy compensator for a nonlinear quad-rotor attitude control system with model uncertainties and actuator fault. The adaptive fuzzy system compensates for the nonlinear function and fault estimation error, and the parallel fuzzy system improves the attitude control speed. For UAV systems with external disturbance, parameter uncertainties, time-variable state delay, and actuator fault, Ref. [17] proposes a sliding mode predictive fault-tolerant control method based on whale optimization algorithm (WOA). The global sliding mode surface is used to establish the prediction model to ensure global robustness, and the WOA is used in the rolling optimization part to accelerate the convergence speed of the method. Aiming at an actuator fault in the aircraft system, Ref. [18] proposes a sliding mode FTC method based upon multi-objective optimization. Ref. [19] forms an actuator fault detection system based on the observer, which can locate the actuator with abnormal behavior by comparing the actual value and observed value of pitch displacement and roll displacement. The adaptive fault-tolerant scheme based on terminal sliding mode control (TSMC) proposed in [20] can make the quad-rotor system with external disturbances, parametric uncertainties, and actuator fault realize finite-time stability. A finite-time exact observer (FEO) is designed to estimate disturbances and reduce the conservatism of disturbances.

However, most of the existing FTC research focuses on actuator faults, and there is relatively little research on sensor faults. Ref. [21] studies nonlinear systems affected by noise and proposes a proportional multiple-integral sliding mode observer (PMISMO) that can simultaneously observe state, actuator, and sensor faults. Finally, the effectiveness of the method is verified in the simulator of an octocopter-type UAV. In [22], the Cubature Kalman filter (CKF) is used to detect and isolate sensor faults. Compared with several nonlinear Kalman filters (KFs), CKF has the smallest estimation error. Then, two active fault-tolerant methods are designed. In [23], the author designs an active FTC method that uses a multi-dimensional generalized fuzzy observer to estimate faults for nonlinear T–S (Takagi–Sugeno) fuzzy systems with cascaded sensor faults, which effectively avoids the coupling effect between faults and enhances the robustness of the system. For discrete switching systems with the sensor fault, Ref. [24] designs a discrete switching generalized observer to solve the problem of simultaneous estimation of the system state and the sensor fault, and finally uses the estimated information to compensate for the failure. However, none of the above references considers time delays.

Actuators, sensors, and communication networks involved in the feedback loop in the control system usually generate delays [25], and such delays usually make it difficult for the controlled variable to respond significantly to external disturbances in a short period of time, which in turn induces a response overshoot. As is known to all, the existence of time delay may make the control system oscillate or reduce the system performance [26]. Therefore, the study of robust control for systems with time delays is also critical in practical applications. For T–S fuzzy systems with an unbounded sensor fault and state delays, Ref. [27] designs a fuzzy augmented state and fault estimation observer for estimating the system state and sensor fault at the same time. Then, a dynamic output feedback controller is designed to compensate for the sensor fault based on the estimated information. This

paper has not considered input delays. In [28], an adaptive memory-free state feedback fault-tolerant controller is designed for a class of systems with actuator fault, multiple time-varying state delays, mismatched parameter uncertainties, and external disturbances. The proposed method makes the tracking error asymptotically converge to zero and the dynamic signal robust tracking effect is better. For a type of continuous system with multi-delay and sensor fault, Ref. [29] first designs an observer for estimating both sensor fault and system state and then designs a controller to handle the fault based on the observer. This paper successfully extends the existing methods to systems with multiple time delays.

According to our knowledge, there are rarely related research on discrete uncertain quad-rotor UAV systems with state delay, input delay, and sensor fault. Aiming at this and comparing it with [30,31], this paper proposes a sliding mode predictive fault-tolerant control method based on the DIECOA. The innovations and main contributions are as follows:

1.  The above two papers deal with the actuator fault of the system, and this paper deals with the sensor fault. In the system model, Ref. [31] has not considered time delays and the parameter uncertainties of the system, and Ref. [30] only considered state time delay. First, the augmented system is constructed in system structural transformation, and the sensor fault is added to the system state.
2.  About the design of sliding mode controller, different from the linear sliding surface in the predictive model [30] and the delta operator approach [31], the quasi-integral sliding mode surface is adopted in this paper to deal with discrete sliding mode control problems which can eliminate the sliding mode approach process and ensure the global robustness of the system.
3.  For the sensor fault, various disturbances, and multiple time delays, this paper designs an intelligent double-power function reference trajectory considering fault and disturbance compensation term, which effectively reduces the adverse effect of multi-delay and improves the precision of fault-tolerant control.
4.  Regarding the design of rolling optimization, which is what [31] lacks, this paper adopts the dynamic information exchange coyote optimization algorithm (DIECOA), which introduced a dynamic information exchange strategy (DIE) to further improve the individual change mechanism of the coyote optimization algorithm (COA). Compared with the multi-agent particle swarm optimization algorithm (MAPSO) proposed in [31], DIECOE improves the local solution capability of the control law optimization in this paper and has the advantages of faster convergence and higher accuracy.

The remaining sections of this article are arranged as follows. In Section 2, the mathematical model of the quad-rotor UAV is established. In Section 3, the fault-tolerant control method is designed. Firstly, a sliding mode prediction model of the quasi-integral type is established. Then, the reference trajectory of the double power function with fault and disturbance compensation is designed. In the rolling optimization part, the DIECOA is designed. In Section 4, the stability of the method is proved. In Section 5, simulation experiments are carried out to illustrate the practicability of the method, and further comparative experiments verify the superiority of the method. Section 6 summarizes the whole paper.

## 2. Problem Statement and Preliminaries

Consider the following discrete system with multi-delay, parameter uncertainties, and external disturbance:

$$\begin{cases} x(k+1) = (A + \Delta A)x(k) + (A_d + \Delta A_d)x(k - \tau_1) + \\ (B + \Delta B)\delta[x(k), u(k)] + (B_d + \Delta B_d)\delta[x(k), u(k - \tau_2)] + \vartheta(k) \\ y(k) = Cx(k) \end{cases} \tag{1}$$

where $x(k) \in R^n$, $u(k) \in R^p$, $y(k) \in R^q$, $\vartheta(k) \in R^n$, respectively, are state, input, output, and external disturbance; $\delta$ is nonlinear function. $A \in R^{n \times n}$, $B \in R^{n \times p}$, $A_d \in R^{n \times n}$, $B_d \in R^{n \times p}$, $C \in R^{q \times n}$ are constant matrices, $\Delta A$, $\Delta A_d$, $\Delta B$, $\Delta B_d$ are parameter uncertainties

of the system, $\tau_1 \in R^+$, $\tau_2 \in R^+$ are respectively state and input time delays, and they both have upper bounds $\tau_{1up}$, $\tau_{2up}$.

Comment: For the selection of parameter uncertainty, please refer to [17].

The quad-rotor UAV has the characteristics of high real-time performance, and when the motor is in a high-frequency working state for a long time, it will cause the body to shake, which will cause the deviation of the accelerometer (sensor). Commonly used accelerometers include capacitive accelerometers, which can measure the value of acceleration through the output voltage. When the UAV is displaced, the capacitance between the sensors will change. If the change in the sensor's output voltage is measured, it is equivalent to measuring the displacement of the UAV. Therefore, the fault of the accelerometer affects the displacement (flight trajectory) of the UAV. According to the fault causes, sensor faults can be divided into stuck faults, bias faults, loss of effectiveness, periodic interference faults, etc. According to the fault modeling angle, sensor faults can be divided into additive faults and multiplicative faults. From the modeling point of view, the sensor additive fault is studied in this paper.

When the sensor of the system fails, the system model changes into the following form:

$$
\begin{cases}
x(k+1) = (A + \Delta A)x(k) + (A_d + \Delta A_d)x(k - \tau_1) + \\
(B + \Delta B)\delta[x(k), u(k)] + (B_d + \Delta B_d)\delta[x(k), u(k - \tau_2)] + \vartheta(k) \\
y(k) = Cx(k) + Df_s(k)
\end{cases}
\tag{2}
$$

$D \in R^{q \times m}$ is a constant matrix, and $f_s(k) \in R^m$ is an additive fault function. System (2) can be rewritten as follows:

$$
\begin{cases}
x(k+1) = Ax(k) + A_d x(k - \tau_1) + \\
B\delta[x(k), u(k)] + B_d \delta[x(k), u(k - \tau_2)] + \omega(k) \\
y(k) = Cx(k) + Df_s(k)
\end{cases}
\tag{3}
$$

where $\omega(k) = \Delta A x(k) + \Delta A_d x(k - \tau_1) + \Delta B \delta[x(k), u(k)] + \Delta B_d \delta[x(k), u(k - \tau_2)] + \vartheta(k)$ represents the sum of system uncertainties and external disturbance. The following assumptions will help obtain the results of this article.

**Assumption 1.** *The function $\delta : R^{n \times m} \to R^r$ satisfies Lipschitz condition for $x(k)$, that is, for any $x_1(k), x_2(k) \in R^n$ and any $u(k) \in R^m$, the existence of Lipschitz constant $\chi > 0$ enables: $\|\delta[x_1(k), u(k)] - \delta[x_2(k) - u(k)]\| \leq \chi \|x_1(k) - x_2(k)\|$.*

*The Stirling interpolation formula is used to perform linear approximation on the nonlinear term in Equation (3). First, define the following function:*

$$
\chi\delta(x) = \delta\left(x + \frac{\iota}{2}\right) - \delta\left(x - \frac{\iota}{2}\right), \mu\delta(x) = \frac{\delta\left(x + \frac{\iota}{2}\right) + \delta\left(x - \frac{\iota}{2}\right)}{2}
\tag{4}
$$

*where $\delta()$ is a nonlinear function, $\iota \in (0,1)$ is an adjustable real number, $\chi$ is a differential operator, and $\mu$ is an average operator. Let $p\iota = \tilde{x} = x - \bar{x}$, and the Stirling interpolation formula is used to expand $\delta(x)$ at $x = \bar{x}$, and its first-order terms are retained, its high-order terms are ignored, and then we can obtain:*

$$
\delta(x) \approx \delta(\bar{x}) + \delta_\iota(\bar{x}) p\iota = \delta(\bar{x}) + \delta_\iota(\bar{x})\tilde{x}
\tag{5}
$$

*where $\delta_\iota(\bar{x}) = \frac{\delta(\bar{x}+\iota) - \delta(\bar{x}-\iota)}{2\iota}$; then, we expand Equation (5) to a vector form as follows:*

$$
\delta(x) = \delta(\bar{x}) + \tilde{I}_{\sigma x}\delta + \frac{1}{2!}\tilde{I}_{\sigma x}^2\delta + \cdots \approx \delta(\bar{x}) + \tilde{I}_{\sigma x}\delta
\tag{6}
$$

*where $\tilde{I}_{\sigma x}\delta = \frac{1}{\iota}\left(\sum_{i=1}^{n}\Delta x_i \mu_i \varepsilon_i\right)\delta(\bar{x})$. Therefore, the Stirling interpolation formula is used to expand $\delta[x(k), u(k)]$ and $\delta[x(k), u(k - \tau_2)]$ in system (3) at $(x_0, u_0)$ and $(x_0, u_0)$, respectively, we can obtain:*

$$\delta(x(k), u(k)) \approx \delta(x_0, u_0) + \frac{1}{\iota}\mu_1\chi_1(x(k) - x_0)\delta(x_0, u_0) + \frac{1}{\iota}\mu_2\chi_2(u(k) - u_0)\delta(x_0, u_0) \quad (7)$$

$$\delta(x(k), u(k)) \approx \delta(x_0, u_0) + I_1'(x(k) - x_0) + I_2'(u(k - \tau_2) - u_0) \quad (8)$$

*Define $I_1 = \frac{\chi_1\mu_1\delta(x_0, u_0)}{\iota}$, $I_2 = \frac{\chi_2\mu_2\delta(x_0, u_0)}{\iota}$, $I_1' = \frac{\chi_1'\mu_1'\delta(x_0, u_0)}{\iota}$, $I_2' = \frac{\chi_2'\mu_2'\delta(x_0, u_0)}{\iota}$; then, Equations (7) and (8) can be rewritten as (9) and (10):*

$$\delta(x(k), u(k)) \approx \delta(x_0, u_0) + I_1(x(k) - x_0) + I_2(u(k) - u_0) \quad (9)$$

$$\delta(x(k), u(k)) \approx \delta(x_0, u_0) + I_1'(x(k) - x_0) + I_2'(u(k - \tau_2) - u_0) \quad (10)$$

*Therefore, the nonlinear terms (9) and (10) are approximate to (11) and (12), respectively:*

$$\delta(x(k), u(k)) \approx I_1 x(k) + I_2 u(k) + I_0 \quad (11)$$

$$\delta(x(k), u(k - \tau_2)) \approx I_1' x(k) + I_2' u(k - \tau_2) + I_0' \quad (12)$$

*where $I_0 = \delta(x_0, u_0) - I_1 x_0 - I_2 u_0$, $I_0' = \delta(x_0, u_0) - I_1' x_0 - I_2' u_0$, then system (3) can be approximate to the linear system (13):*

$$\begin{cases} x(k + 1) = Ax(k) + A_d x(k - \tau_1) + \\ B[I_1 x(k) + I_2 u(k) + I_0] + B_d[I_1' x(k) + I_2' u(k - \tau_2) + I_0'] + \omega(k) \\ y(k) = Cx(k) + Df_s(k) \end{cases} \quad (13)$$

*where $A_p = A + BI_1 + B_d I_1'$, $B_p = BI_2$, $B_p' = BI_2'$, $\omega_p(k) = \omega(k) + BI_0 + BI_0'$, then we can obtain the following system (14):*

$$\begin{cases} x(k + 1) = A_p x(k) + A_d x(k - \tau_1) + B_p u(k) + B_p' u(k - \tau_2) + \omega_p(k) \\ y(k) = Cx(k) + Df_s(k) \end{cases} \quad (14)$$

**Assumption 2.** *$\omega_p(k) = \omega(k) + BI_0 + BI_0'$ has upper and lower bounds, and the rate of change is bounded:*

$$\omega_{\min} \le |\omega_p(k)| \le \omega_{\max} \quad (15)$$

$$|\omega_p(k + 1) - \omega_p(k)| \le \omega_0 \quad (16)$$

*Make a system transformation for system (14), then let: $\bar{x}(k) = \begin{bmatrix} x(k) \\ f_s(k) \end{bmatrix}$, $\bar{x}(k - \tau_1) = \begin{bmatrix} x(k - \tau_1) \\ f_s(k) \end{bmatrix}$, $\bar{\omega}_p(k) = \begin{bmatrix} \omega_p(k) \\ 0 \end{bmatrix}$, $\bar{y}(k) = y(k)$, $\bar{A} = \begin{bmatrix} A_p & 0 \\ 0 & \ell_1 \end{bmatrix}$, $\bar{B} = \begin{bmatrix} B_p & 0 \\ 0 & \frac{1}{2}I_m \end{bmatrix}$, $\bar{A}_d = \begin{bmatrix} A_d & 0 \\ 0 & \ell_2 \end{bmatrix}$, $\bar{B}_d = \begin{bmatrix} B_p' & 0 \\ 0 & \frac{1}{2}I_m \end{bmatrix}$, $\bar{C} = \begin{bmatrix} C & D \end{bmatrix}$, and let $\theta(k) = f_s(k + 1) - (\ell_1 + \ell_2)f(k)$, where $\ell_1, \ell_2$ is new state degrees of freedom. System (14) can be equivalently written as follows:*

$$\begin{cases} \bar{x}(k + 1) = \bar{A}\bar{x}(k) + \bar{A}_d\bar{x}(k - \tau_1) + \bar{B}\begin{bmatrix} u(k) + K_Y y(k) \\ 0 \end{bmatrix} \\ \quad + \bar{B}_d\begin{bmatrix} u(k - \tau_2) + K_Y y(k - \tau_2) \\ 0 \end{bmatrix} + \theta(k)\begin{bmatrix} 0 \\ I_m \end{bmatrix} + \bar{\omega}_p(k) \\ \bar{y}(k) = \bar{C}\bar{x}(k) \end{cases} \quad (17)$$

Let $\bar{u}(k) = \begin{bmatrix} u(k) + K_Y y(k) \\ 0 \end{bmatrix}$, $\bar{u}(k - \tau_2) = \begin{bmatrix} u(k - \tau_2) + K_Y y(k - \tau_2) \\ 0 \end{bmatrix}$, $\bar{\bar{\xi}}(k) = \bar{\omega}_p(k) + \bar{D}\theta(k)$, $\bar{D} = \begin{bmatrix} 0 \\ I_m \end{bmatrix}$.

System (17) is simplified as follows:

$$\begin{cases} \bar{x}(k+1) = \bar{A}\bar{x}(k) + \bar{A}_d\bar{x}(k - \tau_1) + \bar{B}\bar{u}(k) + \bar{B}_d\bar{u}(k - \tau_2) + \bar{\bar{\xi}}(k) \\ \bar{y}(k) = \bar{C}\bar{x}(k) \end{cases} \tag{18}$$

The one-step delay estimation method is used for the estimation of $\bar{\bar{\xi}}(k)$, and let the estimate error be $\hat{\bar{\xi}}(k)$:

$$\hat{\bar{\xi}}(k) = \bar{\bar{\xi}}(k-1) = \bar{x}(k) - \bar{A}x(k-1) - \bar{A}_d\bar{x}(k - \tau_1 - 1) - \bar{B}_d\bar{u}(k - \tau_2 - 1) - B_d\bar{u}(k-1) \tag{19}$$

$$\tilde{\bar{\xi}}(k) = \hat{\bar{\xi}}(k) - \bar{\bar{\xi}}(k) = \bar{\bar{\xi}}(k-1) - \bar{\bar{\xi}}(k) \tag{20}$$

**Assumption 3.** *System faults and uncertainties $\bar{\bar{\xi}}(k)$ are bounded: $\bar{\bar{\xi}}_L \leq \left| \bar{\bar{\xi}}(k) \right| \leq \bar{\bar{\xi}}_U$.*

**Assumption 4.** *The rate of change of system faults and uncertainties are bounded: $\left| \bar{\bar{\xi}}(k) - \bar{\bar{\xi}}(k-1) \right| \leq \bar{\bar{\xi}}_0$.*

**Lemma 1** (Schur's Complement Theorem). *For a given symmetric matrix $\begin{bmatrix} \Lambda_{11} & \Lambda_{12} \\ \Lambda_{21} & \Lambda_{22} \end{bmatrix} < 0$, where $\Lambda_{11}^T = \Lambda_{11}, \Lambda_{22}^T = \Lambda_{22}, \Lambda_{12}^T = \Lambda_{21}$. Then, the above equation is equivalent to 1) $\Lambda_{11} < 0, \Lambda_{21}\Lambda_{11}^{-1}\Lambda_{12} < 0$; 2)$\Lambda_{22} < 0, \Lambda_{12}\Lambda_{22}^{-1}\Lambda_{21} < 0$.*

## 3. SMP-FTC Method Design

### 3.1. SMP Model Analysis

The design of the quasi-integral switching function is as follows. Compared with the linear sliding surface used in [30], the quasi-integral sliding surface can make the system state be located on the sliding surface from the beginning, eliminating the approaching stage of sliding mode and better maintaining the overall robustness. This paper studies the discrete sliding mode control problem. In the discrete sliding mode control system, the state trajectory of the system only moves in the neighborhood of the switching surface, forming a kind of quasi-sliding mode. Therefore, the current research results of continuous system sliding mode control cannot be simply extended to discrete systems. The merits of the quasi-integral sliding surface are that the state trajectory of the system is located on the sliding surface from the initial moment, thereby eliminating the approaching process of the sliding surface. Therefore, the robustness of the system in the whole space is better guaranteed:

$$\begin{cases} s(k) = \Re\bar{x}(k) + \aleph(k) - \Re\bar{x}(0) \\ \aleph(k+1) - \aleph(k) = \Re\bar{x}(k) - \Re\bar{A}\bar{x}(k) - \Re\bar{A}_d\bar{x}(k - \tau_1) \end{cases} \tag{21}$$

where $\aleph(0) = 0$, $\aleph \in R^{p \times n}$ represents a constant matrix, which satisfies that $\Re B$ is nonsingular. We can obtain the following predicted output at the moment $(k + P)$ according to (18) and (21):

$$\begin{aligned} s(k+P) = \Re\Bigg[ &\bar{A}^P\bar{x}(k) + \sum_{i=1}^{P} \bar{A}^{i-1}\bar{A}_d\bar{x}(k+P-i-\tau_1) \\ &+ \sum_{i=1}^{M-1} \bar{A}^{P-i}\bar{B}\bar{u}(k+i-1) + \sum_{i=1}^{P-M} \bar{A}^i\bar{B}\bar{u}(k+M-1) \\ &+ \sum_{i=1}^{M+\tau_2(k)-1} \bar{A}^{P-i}\bar{B}_d\bar{u}(k+i-1-\tau_2) \\ &+ \sum_{i=1}^{P-M-\tau_2(k)} \bar{A}^i\bar{B}_d\bar{u}(k+M-1)\Bigg] + \aleph(k+P) - \Re\bar{x}(0) \end{aligned} \tag{22}$$

where $P$ represents the prediction time horizon, and $M$ represents the control time horizon. The vector form of (22) is as follows:

$$S_{PM}(k) = \Gamma X_d(k) + \Xi X(k) + Y U_d(k) + \Omega U(k) + \Theta(k) \tag{23}$$

where

$S_{PM}(k) = [s(k+1), s(k+2), ..., s(k+P)]^T;$

$\Xi = \left[ (\Re\bar{A})^T, (\Re\bar{A}^2)^T, ..., (\Re\bar{A}^P)^T \right]^T;$

$X_d(k) = [\bar{x}(k-\tau_1(k)), \bar{x}(k+1-\tau_1(k+1)), ..., \bar{x}(k+P-1-\tau_1(k+P-1))]^T;$

$X(k) = [\bar{x}(k+1), ..., \bar{x}(k+P)]^T;$

$U_d(k) = [\bar{u}(k-\tau_2(k)), \bar{u}(k+1-\tau_2(k+1)), ..., \bar{u}(k+M-1)]^T;$

$U(k) = [\bar{u}(k), \bar{u}(k+1), ..., \bar{u}(k+M-1)]^T;$

$\Theta(k) = [\aleph(k+1) - \Re\bar{x}(0), \aleph(k+2) - \Re\bar{x}(0), ..., \aleph(k+P) - \Re\bar{x}(0)]^T;$

$$\Gamma = \begin{bmatrix} \Re\bar{A}_d & 0 & \cdots & \cdots & 0 \\ \Re\bar{A}\bar{A}_d & \Re\bar{A}_d & & & 0 \\ \vdots & \vdots & \ddots & & \vdots \\ \vdots & \vdots & & \ddots & \vdots \\ \Re\bar{A}^{P-1}\bar{A}_d & \Re\bar{A}^{P-2}\bar{A}_d & \cdots & \cdots & \Re\bar{A}_d \end{bmatrix};$$

$$Y = \begin{bmatrix} \Re\bar{B}_d & 0 & \cdots & \cdots & 0 \\ \Re\bar{A}\bar{B}_d & \Re\bar{B}_d & 0 & \cdots & 0 \\ \vdots & \vdots & \cdots & \cdots & \vdots \\ \Re\bar{A}^{M-1}\bar{B}_d & \Re\bar{A}^{M-2}\bar{B}_d & \cdots & \Re\bar{A}\bar{B}_d & \Re\bar{B}_d \\ \Re\bar{A}^M\bar{B}_d & \Re\bar{A}^{M-1}\bar{B}_d & \cdots & \Re\bar{A}^2\bar{B}_d & \Re\bar{A}\bar{B}_d + \Re\bar{B}_d \\ \vdots & \vdots & \cdots & \vdots & \vdots \\ \Re\bar{A}^{P-1}\bar{B}_d & \Re\bar{A}^{P-2}\bar{B}_d & \cdots & \Re\bar{A}^{P-M+1}\bar{B}_d & \sum_{i=0}^{P-M}\Re\bar{A}^i\bar{B}_d \end{bmatrix};$$

$$\Omega = \begin{bmatrix} \Re\bar{B} & 0 & \cdots & \cdots & 0 \\ \Re\bar{A}\bar{B} & \Re\bar{B} & 0 & \cdots & 0 \\ \vdots & \vdots & \cdots & \cdots & \vdots \\ \Re\bar{A}^{M-1}\bar{B} & \Re\bar{A}^{M-2}\bar{B} & \cdots & \Re\bar{A}\bar{B} & \Re\bar{B} \\ \Re\bar{A}^M\bar{B} & \Re\bar{A}^{M-1}\bar{B} & \cdots & \Re\bar{A}^2\bar{B} & \Re\bar{A}\bar{B} + \Re\bar{B} \\ \vdots & \vdots & \cdots & \vdots & \vdots \\ \Re\bar{A}^{P-1}\bar{B} & \Re\bar{A}^{P-2}\bar{B} & \cdots & \Re\bar{A}^{P-M+1}\bar{B} & \sum_{i=0}^{P-M}\Re\bar{A}^i\bar{B} \end{bmatrix}.$$

*3.2. Stability Analysis of the SMP Model*

According to $s(k+1) = s(k) = 0$, the corresponding equivalent control law can be derived as (24):

$$u_{eq}(k) = -(\Re\bar{B})^{-1}\Re\bar{\xi}(k) = -(\Re\bar{B})^{-1}\Re[\bar{\omega}_p(k) + \bar{D}\theta(k)] \tag{24}$$

Substitute (24) into system (18), the ideal sliding mode equation can be obtained as (25):

$$\bar{x}(k+1) = \bar{A}\bar{x}(k) + \bar{A}_d\bar{x}(k-\tau_1) + \bar{B}_d\bar{u}(k-\tau_2) + \left[I - B(\Re B)^{-1}\Re\right]\bar{\xi}(k) \tag{25}$$

**Theorem 1.** *For the system (18), the quasi-integral SMP model is determined by (21), if there is a positive definite matrix* $H_i, (i = 1, 2, 3)$, *which satisfies the inequality (26), then the Equation (25) is globally asymptotically stable:*

$$
\begin{bmatrix}
\Delta_1 & 0 & 0 & 0 & 0 \\
* & \Delta_2 & 0 & 0 & 0 \\
* & * & \Delta_3 & 0 & 0 \\
* & * & * & 4H_1 & \sqrt{2}H_1\bar{B} \\
* & * & * & * & -\bar{B}^T H_1 \bar{B}
\end{bmatrix} < 0 \tag{26}
$$

*where* $\Delta_1 = 5\bar{A}^T H_1 \bar{A} - H_1 + H_2$, $\Delta_2 = 4\bar{A}_d^T H_1 \bar{A}_d + \bar{A}_d^T H_2 \bar{A}_d - H_2$, $\Delta_3 = 4\bar{B}_d^T H_1 \bar{B}_d + \bar{A}_d^T H_2 \bar{A}_d - H_3$.

**Proof.** Choose the Lyapunov function (27) for Equation (25):

$$
V(k) = \bar{x}^T(k)H_1\bar{x}(k) + \sum_{i=k-\tau_1(k)}^{k-1} \bar{x}^T(i)H_2\bar{x}(i) + \sum_{j=k-\tau_2(k)}^{k-1} \bar{u}^T(j)H_3\bar{u}(j) \tag{27}
$$

Choose $\Re = \bar{B}^T H_1$, which can guarantee that $\Re B$ is nonsingular, and let $Q = (\bar{B}^T H_1 \bar{B})^{-1} \bar{B}^T H_1$.

The difference equation of the Lyapunov function along the state trajectory of the ideal sliding mode (25) satisfies:

$$
\begin{aligned}
\Delta V(k) &= V(k+1) - V(k) \\
&= \bar{x}^T(k+1)H_1\bar{x}(k+1) + \sum_{i=k+1-\tau_1(k+1)}^{k} \bar{x}^T(i)H_2\bar{x}(i) - \bar{x}^T(k)H_1\bar{x}(k) \\
&\quad + \sum_{j=k+1-\tau_2(k+1)}^{k} \bar{u}^T(j)H_3\bar{u}(j) - \sum_{i=k-\tau_1(k)}^{k-1} \bar{x}^T(i)H_2\bar{x}(i) - \sum_{j=k-\tau_2(k)}^{k-1} \bar{u}^T(j)H_3\bar{u}(j) \\
&= \bar{x}^T(k)\bar{A}^T H_1 \bar{A}\bar{x}(k) + \bar{x}^T(k)(H_2 - H_1)\bar{x}(k) + 2\bar{x}^T(k)\bar{A}^T H_1 \bar{A}_d\bar{x}(k-\tau_1) \\
&\quad + 2\bar{x}^T(k)\bar{A}^T H_1 \bar{B}_d\bar{u}(k-\tau_2) + \bar{x}^T(k-\tau_1)\bar{A}_d^T H_1 \bar{A}_d\bar{x}(k-\tau_1) - \bar{x}^T(k-\tau_1)H_2\bar{x}(k-\tau_1) \\
&\quad + 2\bar{x}^T(k-\tau_1)\bar{A}_d^T H_1 \bar{B}_d\bar{u}(k-\tau_2) + \bar{u}^T(k-\tau_2)\bar{B}_d^T H_1 \bar{B}_d\bar{u}(k-\tau_2) \\
&\quad - \bar{u}^T(k-\tau_2)H_3\bar{u}(k-\tau_2) + 2\bar{x}^T(k)\bar{A}^T H_1\bar{\xi}(k) - 2\bar{x}^T(k)\bar{A}^T H_1\bar{B}Q\bar{\xi}(k) \\
&\quad + 2\bar{x}^T(k-\tau_1)\bar{A}_d^T H_1\bar{\xi}(k) - 2\bar{x}^T(k-\tau_1)\bar{A}_d^T H_1\bar{B}Q\bar{\xi}(k) + 2\bar{u}^T(k-\tau_2)\bar{B}_d^T H_1\bar{\xi}(k) \\
&\quad - 2\bar{u}^T(k-\tau_2)\bar{B}_d^T H_1\bar{B}Q\bar{\xi}(k) + \bar{\xi}^T(k)H_1\bar{\xi}(k) - \bar{\xi}^T(k)H_1\bar{B}Q\bar{\xi}(k) \\
&\leq 5\bar{x}^T(k)\bar{A}^T H_1 \bar{A}\bar{x}(k) + \bar{x}^T(k)(H_2 - H_1)\bar{x}(k) + 4\bar{x}^T(k-\tau_1)\bar{A}_d^T H_1 \bar{A}_d\bar{x}(k-\tau_1) \\
&\quad + \bar{x}^T(k-\tau_1)\bar{A}_d^T H_2 \bar{A}_d\bar{x}(k-\tau_1) - \bar{x}^T(k-\tau_1)H_2\bar{x}(k-\tau_1) \\
&\quad + 4\bar{u}^T(k-\tau_2)\bar{B}_d^T H_1 \bar{B}_d\bar{u}(k-\tau_2) + \bar{u}^T(k-\tau_2)\bar{B}_d^T H_2 \bar{B}_d\bar{u}(k-\tau_2) \\
&\quad - \bar{u}^T(k-\tau_2)H_3\bar{u}(k-\tau_2) + 4\bar{\xi}^T(k)H_1\bar{\xi}(k) + 2\bar{\xi}^T(k)H_1\bar{B}Q\bar{\xi}(k) \\
&= \begin{bmatrix} \bar{x}(k) & \bar{x}(k-\tau_1) & \bar{u}(k-\tau_2) & \bar{\xi}(k) \end{bmatrix} \Phi \begin{bmatrix} \bar{x}(k) \\ \bar{x}(k-\tau_1) \\ \bar{u}(k-\tau_2) \\ \bar{\xi}(k) \end{bmatrix}^T
\end{aligned} \tag{28}
$$

where $\Phi = \begin{bmatrix} \Delta_1 & & & \\ & \Delta_2 & & \\ & & \Delta_3 & \\ & & & \Delta_4 \end{bmatrix}$, $\Delta_4 = 4H_1 + 2H_1 BQ$.

From $\Phi < 0$, $\Delta V(k) < 0$ can be obtained. From Lemma 1, $\Phi < 0$ is equivalent to (26). Therefore, when the inequality (26) holds, the ideal sliding mode (25) is asymptotically stable. The stability of prediction model is demonstrated. □

### 3.3. Feedback Correction

$s(k|k - P)$ is defined as the predictive output of the switching function at the moment $s(k|k - P)$, and $e_s(k)$ is defined as the predictive error at the moment $k$:

$$
\begin{aligned}
s(k|k - P) = \Re\Bigg[ &\bar{A}^P\bar{x}(k - P) + \sum_{i=1}^{P} \bar{A}^{i-1}\bar{A}_d\bar{x}(k - i - \tau_1) \\
&+ \sum_{i=1}^{M-1} \bar{A}^{P-i}\bar{B}\bar{u}(k - P + i - 1) + \sum_{i=1}^{P-M} \bar{A}^i\bar{B}\bar{u}(k - P + M - 1) \\
&+ \sum_{i=1}^{M+\tau_2(k)-1} \bar{A}^{P-i}\bar{B}_d\bar{u}(k - P + i - 1 - \tau_2) \\
&+ \sum_{i=1}^{P-M-\tau_2(k)} \bar{A}^i\bar{B}_d\bar{u}(k - P + M - 1) \Bigg] + \aleph(k) - \Re\bar{x}(0)
\end{aligned}
\tag{29}
$$

$$
e_s(k) = s(k) - s(k|k - P)
\tag{30}
$$

Then, the $P$ step predictive output of the switching function after error correction is as (30), where $f_p$ is the correction coefficient. The vector form of (31) is as (32):

$$
\widetilde{s}(k + P) = s(k + P) + f_p e_s(k)
\tag{31}
$$

$$
\tilde{S}_{PM}(k) = S_{PM}(k) + F_p E_s(k)
\tag{32}
$$

where

$$
\widetilde{S}_{PM}(k) = [\widetilde{s}(k + 1), \widetilde{s}(k + 2), \dots, \widetilde{s}(k + P)]^T;
$$

$$
F_P = \begin{bmatrix} f_1 & & & \\ & f_2 & & \\ & & \ddots & \\ & & & f_P \end{bmatrix}, 1 \geq f_1 \geq f_2 \geq \cdots \geq f_P > 0;
$$

$$
E_S(k) = [s(k) - s(k|k - 1), s(k) - s(k|k - 2), \dots, s(k) - s(k|k - P)]^T.
$$

### 3.4. Reference Trajectory Design

Considering that the system has state delay and input delay, this section designs the reference trajectory of the double power function. Compared with the reference trajectory of [30], the dynamic convergence speed of the double-power reference trajectory is fast, and the accuracy is high. The system has sensor fault, parameter uncertainty, and external disturbances, so the improved fault and disturbance compensation is designed, which effectively handles faults and various disturbances. First, the design of the improved fault and disturbance compensation is based on the traditional boundary layer chattering suppression strategy. Then, the equivalent fault and disturbance rate of change is defined as the second-order difference of the fault and disturbance. As a result, the quasi-sliding mode bandwidth is reduced, and chattering is suppressed.

The reference trajectory designed in this paper is as (33):

$$
\begin{cases}
s_{ref}(k + 1) = (1 - qT)s_{ref}(k) - \varepsilon_1 T\left|s_{ref}(k)\right|^{\alpha} - \varepsilon_2 T\left|s_{ref}(k)\right|^{\beta}\text{sgn}\left(s_{ref}(k)\right) \\
\quad + (1 - z^{-1})\Re\left[\bar{\xi}(k) - \bar{\xi}(k - 1)\right] \\
s_{ref}(k) = s(k)
\end{cases}
\tag{33}
$$

where $\varepsilon_1 > 0, \varepsilon_2 > 0, q > 0, 1 - qT > 0, 0 < \alpha < 1, \beta > 1$, $T$ represents sampling time. Simplify to obtain the reference trajectory as (34):

$$
\begin{cases}
s_{ref}(k + 1) = (1 - qT)s_{ref}(k) - \varepsilon_1 T\left|s_{ref}(k)\right|^{\alpha} - \varepsilon_2 T\left|s_{ref}(k)\right|^{\beta}\text{sgn}\left(s_{ref}(k)\right) \\
\quad + \Im(k) \\
s_{ref}(k) = s(k)
\end{cases}
\tag{34}
$$

$$\Im(k) = \Re\left[\bar{\bar{\xi}}(k) - 2\bar{\bar{\xi}}(k-1) + \bar{\bar{\xi}}(k-2)\right] \tag{35}$$

**Lemma 2.** *From [32], when the zero-order holder is used for discretization, the equivalent fault and disturbance $\bar{\bar{\xi}}(k)$ in the system (18) have the property as: $\bar{\bar{\xi}}(k) - 2\bar{\bar{\xi}}(k-1) + \bar{\bar{\xi}}(k-2) = O(T^3)$, where $O(T)$ indicates that the magnitude of $\bar{\bar{\xi}}(k)$ is on the order of magnitude $O(T)$.*

**Assumption 5.** *The change rate of equivalent fault and disturbance $\Im(k)$ defined by (35) is bounded, and $|\Im(k)| \le UpB \le \min\{\varepsilon_1, \varepsilon_2\}T$, and $UpB$ represents the upper bound of $\Im(k)$.*

*The traditional change rate of equivalent fault and disturbance $\Im_1(k)$ and its upper bound $UpB_1$ is as (36) [33]:*

$$\begin{cases} \Im_1(k) = \Re\left[\bar{\bar{\xi}}(k) - \bar{\bar{\xi}}(k-1)\right] \\ |\Im_1(k)| \le UpB_1 \end{cases} \tag{36}$$

*3.5. Rolling Optimization Design Based on DIECOA*

At time $k$, the optimization performance indicator is shown in (37), and (38) represents the vector form of (37):

$$j(k) = \sum_{i=1}^{P} \lambda_i \left[s_{ref}(k+i) - \tilde{s}(k+1)\right]^2 + \sum_{l=1}^{M} \rho_l \left[u(k+l-1)\right]^2 \tag{37}$$

$$J(k) = \left[S_{ref}(k) - \tilde{S}_{PM}(k)\right]^T \mathrm{H}_4 \left[S_{ref}(k) - \tilde{S}_{PM}(k)\right] + [U(k)]^T \mathrm{H}_5 [U(k)] \tag{38}$$

where $\lambda_i, \rho_i$ is the non-negative weight,

$$\mathrm{H}_4 = \begin{bmatrix} \lambda_1 & & & \\ & \lambda_2 & & \\ & & \ddots & \\ & & & \lambda_P \end{bmatrix}, \mathrm{H}_5 = \begin{bmatrix} \rho_1 & & & \\ & \rho_2 & & \\ & & \ddots & \\ & & & \rho_M \end{bmatrix}.$$

The Coyote Optimization Algorithm (COA) [34] is a new and promising global optimization algorithm based on random populations. Unlike most intelligent optimization algorithms, the population of COA is subdivided, and each individual is affected by internal social culture. Moreover, the design of COA requires only a few control parameters, such as the number of wolves and population size. However, the traditional COA maintains a constant update mechanism during the solution process, weakening the global search capability; it is easier to fall into premature convergence when solving complex objective functions. Therefore, this paper designs DIECOA.

This paper introduces an improved dynamic information exchange (DIE) strategy to update the individual replacement formula, which effectively strengthens the information interaction and integration of individuals within the population, and also has a positive impact on the growth of new coyote individuals, with higher convergence accuracy and faster speed. Ref. [31] has not yet adopted the optimization algorithm. Compared with the MAPSO algorithm adopted in [30], DIECOA proposed in this paper has higher convergence accuracy and faster speed. First, make $J(k)$ take the minimum value $U(k)$, that is, take the optimized performance (38) as the fitness function of DIECOA.

The implementation process of DIECOA is as follows:

(1) Coyote population initialization

Set parameters: coyote group $N_p$, number of coyotes per group $N_c$, dimension $D$, and termination condition $nfevalMAX$. Randomly initializing the group, Equation (40) represents the social state variable set of the $i$-th individual in the $p$ group at the $t$-th time, and Equation (39) is obtained by assigning the $w$-th dimension to the $i$-th individual in the group $p$ at the $t$-th time in the variable set:

$$y_{c,w}^{p,t} = lb_w + r_w(ub_w - lb_w) \tag{39}$$

$$y_c^{p,t} = \left(y_{c,1}^{p,t}, y_{c,2}^{p,t}, y_c^{p,t}, \ldots, y_{c,D}^{p,t}\right) \tag{40}$$

where $ub_w, lb_w$ respectively denote the upper and lower bounds of the w-th dimension value, and $r_w \in [0,1]$ is a random real number.

(2) Evaluation of coyote adaptability

Calculate the fitness value of a coyote individual:

$$Adapt_i^{p,t} = A\left(y_i^{p,t}\right) \tag{41}$$

(3) Coyote population evolution trend

First, find the leader $Clead^{p,t}$ of the pack, and calculate current cultural trends $cult^{p,t}$ in coyote populations:

$$Clead^{p,t} = \left\{ y_i^{p,t} \middle\| \arg_{\{i=1,2,\ldots,N_c\}} \min A\left(y_i^{p,t}\right) \right\} \tag{42}$$

$$cult_j^{p,t} = \begin{cases} O_{\frac{N_c+1}{2},w}^{p,t}, & N_c \text{ is 'odd.} \\ O_{\frac{N_c}{2},w} + O_{\frac{N_c+1}{2},w}^{p,t}, & \text{others.} \end{cases} \tag{43}$$

where $O_{\frac{N_c+1}{2},w}^{p,t}$ represents the median of the social state of the $w$-th dimension variable of all individuals in the group $p$ at the $t$-th time when $N_c$ is an odd number.

Defining $P_e$ as the probability of coyote group transition, that is, the probability that they actively or passively departure from the original coyote group:

$$P_e = 0.005 N_c^2 \tag{44}$$

Birth and death of individuals: record the age of the coyote (in years) as $year_c^{p,t}$. $\left(pup^{p,t}\right)$ represents a coyote newborn, which is written as a combination of the social status of both parents (selected at random) of the new coyote plus environmental impact:

$$pup_w^{p,t} = \begin{cases} y_{n_1,w}^{p,t}, & rnd_w < P_s or, w = w_1 \\ y_{n_2,w}^{p,t}, & rnd_w \geq P_s + P_a or, w = w_2 \\ R_w, & rnd_w, else. \end{cases} \tag{45}$$

where $n_1$ and $n_2$ are random coyotes from group $p$, $w_1, w_2$ are two random dimensions of the problem, $R_w, rnd_w \in [0,1]$, which represent random numbers generated with uniform probability.

Associated probability $P_a$ and discrete probability $P_s$ can affect the individual richness and cultural diversity of the coyotes. Define $P_a, P_s$ as follows:

$$P_a = \frac{(1 - P_s)}{2}, P_s = \frac{1}{D} \tag{46}$$

$\omega$ represents the inertia weight of the individual social state, meaning: after birth, the newborn's survival is determined according to the value of $Adapt_i^{p,t}$: if there is at least one

individual in the population whose $Adapt_i^{p,t}$ is smaller than that of the newborn, the newborn survives, and the individual with the smallest $Adapt_i^{p,t}$ dies; if there is no individual in the population with $Adapt_i^{p,t}$ smaller than that of the neonate, the neonate dies.

Calculate the influence of the leader $Clead^{p,t}$ and cultural trends $cult^{p,t}$ on the individual update in the group $p$ at the current moment $t$, and denote them respectively as $Inf_1$ and $Inf_2$:

$$Inf_1 = Clead^{p,t} - y_{cr_1}^{p,t}, Inf_2 = cult^{p,t} - y_{cr_2}^{p,t} \tag{47}$$

where the two random coyotes in the current group are denoted as $cr_1$ and $cr_2$.

(4) Update the coyote individuals in each group

A dynamic information exchange strategy (DIE) is introduced to enhance the information integration within the population and then achieve the purpose of promoting individual growth. DIE improves the limitations of the individual replacement mechanism in the group, expanding the information reserve and interaction degree generated by new individuals in the group so that the replacement of individuals is no longer solely affected by the optimal wolf and the current coyote population culture, but, in the process of change, DIE accepts most of the group of other individuals, the information provided by the mutual influence, to improve the diversity of the individual change and growth. From the perspective of the optimization performance of the algorithm, this strategy has a significant effect on improving the local solution ability of the control law optimization.

The new coyote individual $new\_y_i^{p,t}$ is obtained by updating all coyote individuals in the coyote pack, and then retain the optimal coyote $y_i^{p,t+1}$:

$$new\_y_i^{p,t} = y_i^{p,t} + \kappa_1 R_1 + \kappa_2 R_2 \tag{48}$$

$$y_i^{x,t+1} = \begin{cases} new\_y_i^{x,t}, A\left(new\_y_i^{x,t}\right) < A\left(y_i^{x,t}\right) \\ y_i^{x,t}, others \end{cases} \tag{49}$$

$$R_1 = \kappa_1\left(Clead^{p,t} - y_1^{p,t}\right) + (1-\kappa_1)\left(cul^{p,t} - y_2^{p,t}\right) = \kappa_1 Inf_1 + (1-\kappa_1)Inf_2 \tag{50}$$

$$R_2 = y_1^{p,t} - y_2^{p,t} \tag{51}$$

where $\kappa_1, \kappa_2 \in [0,1]$ are random real numbers, which respectively denote the weight of individual coyotes influenced by the leader $Clead^{p,t}$ of the pack and cultural trends of groups. $R_2$ represents information exchange.

(5) Coyote age update

Simulate the entire stages of an individual's growth over time, age-changing the individual.

(6) Judge termination conditions

Termination condition judgment. If it is reached, output the individual social state with the optimal adaptability; otherwise, return to phase (2) to continue.

The flow diagram of SMP method designed in this paper is shown in Figure 1.

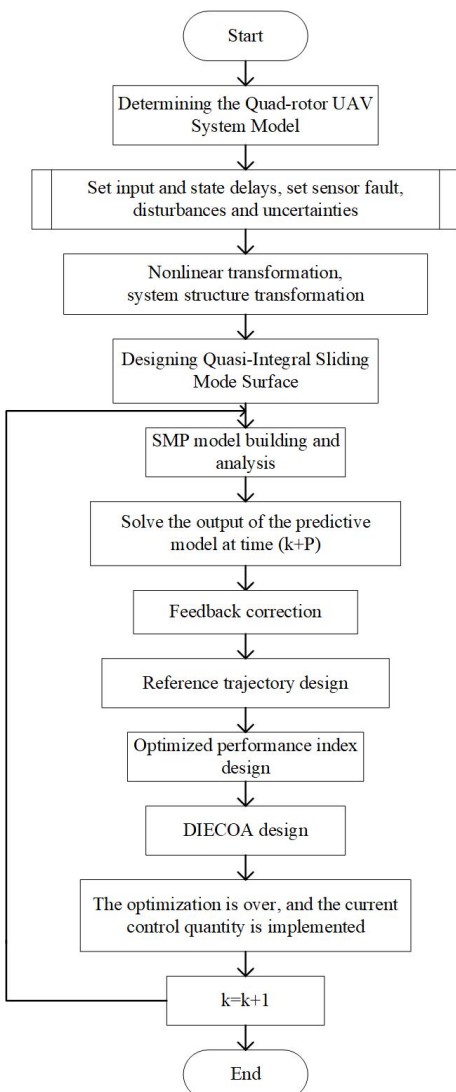

**Figure 1.** Flow diagram of the SMP method.

## 4. Stability Analysis

The overview of the overall closed-loop system is illustrated in Figure 2.

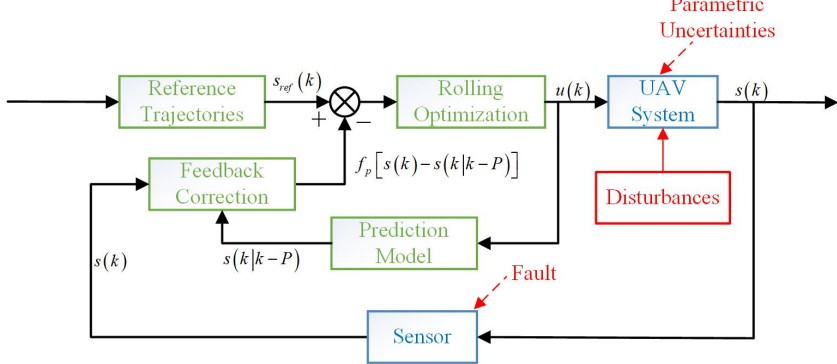

**Figure 2.** Block diagram of the overall closed-loop system.

Making the moment $k$ the current moment, the predicted output at the moment $(k + P)$ of system (18) is as (22), and the actual predicted output vector form is as follows:

$$S_{PM}(k) = \Gamma X_d(k) + \Xi X(k) + Y U_d(k) + \Omega U(k) + \Pi \partial(k) + \Theta(k) \tag{52}$$

$$\text{where } \Pi = \begin{bmatrix} \Re & 0 & \cdots & 0 \\ \Re\bar{A} & \Re & \cdots & 0 \\ \vdots & \vdots & \ddots & \vdots \\ \Re\bar{A}^{P-1} & \Re\bar{A}^{P-2} & \cdots & \Re \end{bmatrix}, \partial(k) = $$

$$\begin{bmatrix} \bar{\xi}(k) & \bar{\xi}(k+1) & \cdots & \bar{\xi}(k+P-1) \end{bmatrix}^T.$$

The optimal control law must satisfy $\frac{\partial J(k)}{\partial U(k)} = 0$; for that, $\frac{\partial J(k)}{\partial U(k)} = 0$ is the necessary condition for $J(k)$ to take the extreme:

$$U(k) = \left(H_5 + \Omega^T H_4 \Omega\right)^{-1} \Omega^T H_4 \Big[ S_{ref}(k) - \Xi X(k) - \Gamma X_d(k) \\ - YU_d(k) - \Theta(k) - F_p E_s(k) \Big] \tag{53}$$

Substitute Equation (53) into Equation (52); then, taking $H_5 = 0$ means that there is no limit to the control input $U(k)$, which is reasonable. $H_4$ take the identity matrix of the corresponding dimension, then:

$$U(k) = \left(H_5 + \Omega^T H_4 \Omega\right)^{-1} \Omega^T H_4 \Big[ S_{ref}(k) - \Xi X(k) - \Gamma X_d(k) \\ - YU_d(k) - \Theta(k) - F_p E_s(k) \Big] \tag{54}$$

$$S_{PM}(k) = S_{ref}(k) + \Pi\partial(k) - F_p E_s(k) \tag{55}$$

In the process of solving the control law with rolling optimization, since the control law is solved in real-time, that is, only the current control input signal is implemented on the controlled object. The correction factor $f_1$ is generally taken as 1, and then we can obtain (56) as follows:

$$\begin{aligned} s(k+1) &= \begin{bmatrix} 1 & 0 & \cdots & 0 \end{bmatrix} S_{PM}(k) \\ &= s_{ref}(k+1) - f_1[s(k) - s(k \mid k-1)] + \Re\bar{\xi}(k) \\ &= s_{ref}(k+1) + \Re\big[\bar{\xi}(k) - f_1\bar{\xi}(k-1)\big] \\ &= s_{ref}(k+1) + \Re\big[\bar{\xi}(k) - \bar{\xi}(k-1)\big] \end{aligned} \tag{56}$$

From Assumption 4: $\left|\bar{\xi}(k) - \bar{\xi}(k-1)\right| \leq \bar{\xi}_0$,

$$s(k+1) = s_{ref}(k+1) + \Re\big[\bar{\xi}(k) - \bar{\xi}(k-1)\big] \leq s_{ref}(k+1) + \Re\bar{\xi}_0 \tag{57}$$

where

$$s_{ref}(k+1) = (1-qT)s_{ref}(k) - \varepsilon_1 T\left|s_{ref}(k)\right|^\alpha - \varepsilon_2 T\left|s_{ref}(k)\right|^\beta \text{sgn}\Big(s_{ref}(k)\Big) + \Im(k). \tag{58}$$

From Assumption 5: $|\Im(k)| \leq UpB$, therefore, we only need to verify the boundedness of the double-power reaching law. Namely, we can only judge the following formula:

$$s_\omega = (1-qT)s_{ref}(k) - \varepsilon_1 T\left|s_{ref}(k)\right|^\alpha - \varepsilon_2 T\left|s_{ref}(k)\right|^\beta \text{sgn}\Big(s_{ref}(k)\Big) \tag{59}$$

**Proof.** Let $\Delta s_\omega(k) = s_\omega(k+1) - s_\omega(k)$, then:

$$\Delta s_\omega(k) = -qTs_\omega(k) - \varepsilon_1 T|s_\omega(k)|^\alpha \text{sgn}(s(k)) - \varepsilon_2 T|s_\omega(k)|^\beta \text{sgn}(s(k)) \tag{60}$$

(1) When $s_\omega(k) \geq 0$:
$\frac{\partial\Delta s_\omega(k)}{\partial s_\omega(k)} = -qT - \varepsilon_1 T\alpha[s_\omega(k)]^{\alpha-1} - \varepsilon_2 T\beta[s_\omega(k)]^{\beta-1}$

From $\varepsilon_1 > 0, \varepsilon_2 > 0, q > 0, 1 - qT > 0, 0 < \alpha < 1, \beta > 1$, we can obtain $\frac{\partial\Delta s_\omega(k)}{\partial s_\omega(k)} < 0$, that is, $\Delta s_\omega(k)$ is a decreasing function of $s_\omega(k)$. When $s_\omega(k) \geq 0$, $\Delta s_\omega(k) \leq -qTs_\omega(k) - \varepsilon_1 T[s_\omega(k)]^\alpha - \varepsilon_2 T[s_\omega(k)]^\beta\Big|_{s_\omega(k)=0} = 0$, and from $\Delta s_\omega(k) \leq 0$, we can obtain that $s_\omega(k)$

decreases until reaching the state $s_\omega(k) = 0$. If and only if $s_\omega(k) = 0$, $\Delta s_\omega(k) = 0$, then $s_\omega(k+1) \leq v$.

(2) When $s_\omega(k) < 0$:

$$\frac{\partial \Delta s_\omega(k)}{\partial s_\omega(k)} = -qT - \varepsilon_1 T\alpha[-s_\omega(k)]^{\alpha-1} - \varepsilon_2 T\beta[-s_\omega(k)]^{\beta-1}$$

The same as (1), we can obtain $\frac{\partial \Delta s_\omega(k)}{\partial s_\omega(k)} < 0$. Namely, $\Delta s_\omega(k)$ is a decreasing function of $s_\omega(k)$. When $s_\omega(k) < 0$, $\Delta s_\omega(k) > -qTs_\omega(k) - \varepsilon_1 T[s_\omega(k)]^{\alpha} - \varepsilon_2 T[s_\omega(k)]^{\beta}\big|_{s_\omega(k)=0} = 0$, and, from $\Delta s_\omega(k) > 0$, we can obtain that $s_\omega(k)$ increases until reaching $s_\omega(k) = 0$. Then, $s_\omega(k+1) \leq v$ is obtained.

(3) When $s_\omega(k) = 0$:

According to $s_\omega(k+1) = s_\omega(k) = 0$, the system is in a stable state. Then, $s_\omega(k+1) \leq v$.

In summary, $s_\omega(k+1) \leq v$ is obtained, and, because $s(k+1) \leq s_\omega(k+1) + UpB + \Re\bar{\xi}_0$, then $|s(k+1)| \leq v + UpB + \Re\bar{\xi}_0$ is obtained. Namely, the closed-loop robust stability of the system is proved. □

## 5. Simulation Settings and Analysis

### 5.1. Model Introduction and Parameter Settings

In this section, we verify the rationality and effectiveness of the proposed method through some simulation examples on the fault-tolerant simulation platform of the quad-rotor UAV. As the name implies, Quad-rotor UAV is composed of four rotors and a rigid airframe. The motion of the UAV is controlled by controlling the speed of the four rotors, and the quality of the control method will directly determine whether the UAV system can maintain robustness when faults or disturbances occur. In this section, the Qball-X4 aircraft produced by QUANSER in Canada is selected as the simulation object, as is shown in Figure 3.

Since the motion of the $x$-axis and $Y$-axis is symmetric, the channel signal of the $x$-axis forward direction is selected as the research object for simulation. The mathematical model of the aircraft body is shown in Table 1 below.

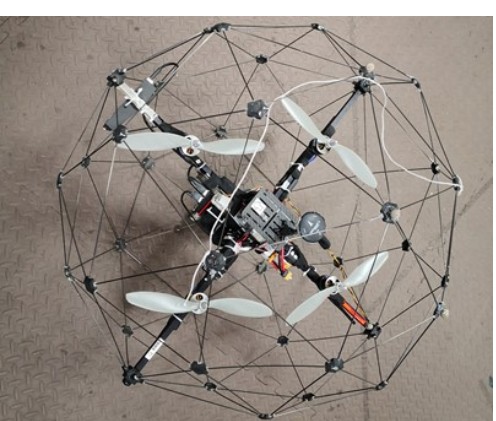

**Figure 3.** Qball-X4 aircraft.

**Table 1.** Mathematic model of Qball-X4.

| Physical Meaning | Expression |
| --- | --- |
| Dynamic equation of X-axis | $M_g \ddot{X} = 4F\sin(\dot{\theta})$ |
| Lift generated by the rotor $F$ | $F = K_g \frac{\omega}{s+\omega} u$ |
| Actuator dynamics $v$ | $v = \frac{\omega}{s+\omega} u$ |
| State space expression form of $v$:$\dot{v}$ | $\dot{v} = -\omega v + \omega u$ |

Where the above mathematical model assumes that the quad-rotor UAV flies at a low speed and a small attitude angle, so the pitch and roll angles are approximately 0,

and considers the influence of lift and pitch angle. *u* represents actuator input. The body parameters of Qball-X4 are shown in Table 2.

**Table 2.** Body parameters of Qball-X4.

| Physical Meaning | Value |
|---|---|
| Body mass | $M_g$ = 1.4 kg |
| The positive gain | $K_g = 120$ N |
| Actuator bandwidth | $\omega = 15$ rad/s |

Let $\sin \theta = \theta$, and the model in the *x*-axis direction can be derived as (61):

$$
\begin{bmatrix} \dot{X} \\ \ddot{X} \\ \dot{v} \end{bmatrix} = \begin{bmatrix} 0 & 1 & 0 \\ 0 & 0 & \frac{4K_g}{M_g}\theta \\ 0 & 0 & -\omega \end{bmatrix} \begin{bmatrix} X \\ \dot{X} \\ v \end{bmatrix} + \begin{bmatrix} 0 \\ 0 \\ \omega \end{bmatrix} u \tag{61}
$$

Due to external conditions such as vibration and power supply 50 Hz interference, the measurement of the sensor is affected by periodic interference. Since periodic interference is one of the more common faults of sensors, a periodic interference fault is taken into consideration in this paper.

In the *x*-axis position control stage, the periodic interference fault is injected into the system model. Considering the parameter uncertainties, external disturbance, the sensor fault, input time delay, and state time delay in the system, the values of the matrix parameters in the quad-rotor UAV system (1) are as follows:

$$
A = \begin{bmatrix} 0 & 1 & 0 \\ 0 & 0 & 12 \\ 0 & 0 & -15 \end{bmatrix}, A_d = \begin{bmatrix} 0 & 0 & 0 \\ 0 & 0 & 4 \\ 0 & 0 & -5 \end{bmatrix}, B = \begin{bmatrix} 0 \\ 0 \\ 15 \end{bmatrix}, B_d = \begin{bmatrix} 0 \\ 0 \\ 1 \end{bmatrix}, C = \begin{bmatrix} 1 & 0 & 0 \end{bmatrix},
$$

constant matrix is $D = \begin{bmatrix} 0.2 & 0.4 & 0.1 \end{bmatrix} \sin(k)$. The parameter uncertainties of the system are $\Delta A = 0.1A$, $\Delta A_d = 0.1A_d$, $\Delta B = 0.1B$, and $\Delta B_d = 0.1B_d$; external disturbance takes the more general white Gaussian noise function, and the sensor fault function is $f_s(k) = \begin{bmatrix} 0.3\sin(6k) \\ 0.2\sin(3k) \\ 0.2\sin(2k) \end{bmatrix}$; sliding mode matrix parameter is $\sigma = \begin{bmatrix} 1 & 1 & 1 \end{bmatrix}$. The PWM wave inputs by the system may have a lag. Therefore, the input time delay needs to be considered. This section takes the input time delay as $\tau_2 = 3$. Then, we consider that the signal transmission in the system is through the wireless network, and the state delay also needs to be considered. The state delay is taken as $\tau_1 = 3$. The prediction time horizon $P$ represents the number of prediction steps that make the prediction output approach the expected value; the prediction time horizon in this paper is taken as $P = 4$. The control time horizon $M$ represents the number of parameters used to obtain future control trajectories. The control time horizon in this paper is taken as $M = 2$. The sampling time and simulation time horizon are selected as $T = 0.02$ and $k = 1000$. All the above matrices have been discretized during the simulation experiment.

The parameter setting of DIECOA: Dimension $D$ is taken as $D = 10$, the maximum number of iterations is set to 50, the coyote group $N_p$ is set to 10, and the number of coyotes per group $N_c$ is set to 10.

*5.2. Simulation Results*

There is a case in which the initial conditions remain the same, that is, in the nonlinear discrete quad-rotor UAV system with multiple time delays, parameter uncertainties, external disturbances, and sensor fault. This section will verify and compare the robustness of the method designed in this paper and the methods in [30,31]. To better illustrate the FTC capability of our method, we evaluate the complete FTC operation by injecting a fault into the system at some point after the simulation starts. It is supposed that the periodic

interference fault occurs on the accelerometer at $k = 200$ and continues throughout the considered scenarios.

From Figures 4 and 5, we can easily see that the stability of the method designed in this paper is obviously better than that of the methods of the comparative references, especially in the position trajectories of the $x$-axis (Figure 4), and we can see that the trajectory of the Qball-x4 aircraft tends to be stable and remains stable when $k = 300$. Moreover, when the methods of [30,31] relatively act on the aircraft, the shaking of the aircraft along the $x$-axis is more severe. When $k = 200$, the sensor fails, and the $x$-axis position of the quad-rotor UAV under the three methods will have a certain offset. As can be seen from Figures 4 and 5, the position offset of the quad-rotor UAV under the action of the method in this paper is more minor, and the fault-tolerant control of the fault can be completed in a faster time.

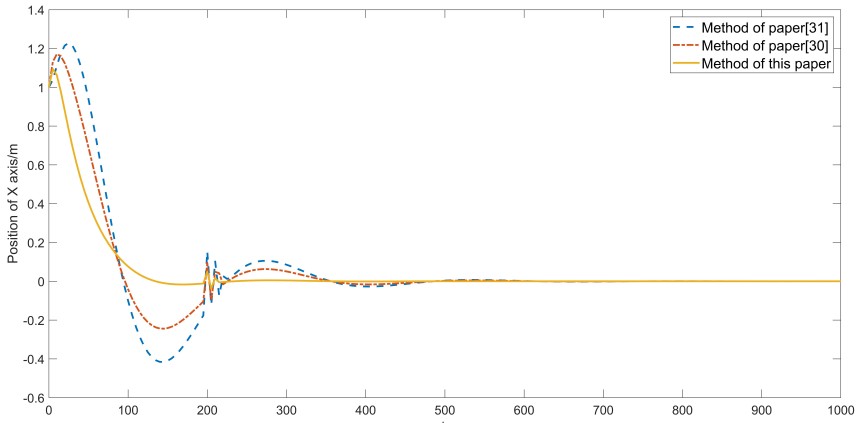

**Figure 4.** the position trajectories of the $x$-axis (1).

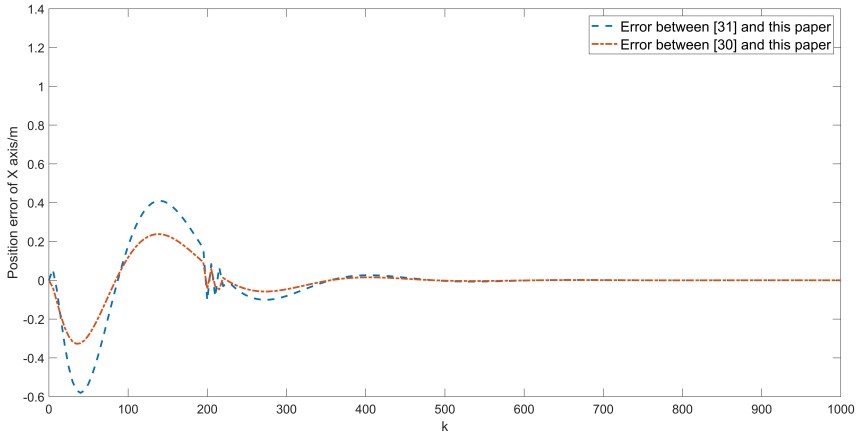

**Figure 5.** the position error of the $x$-axis.

From Figure 6, it can be seen that the method proposed in this paper can stabilize the system faster and weaken chattering. Since the method in [30] only considers the single time delay and the method in [31] has not considered the time delays, the chattering of the systems is relatively more obvious than the method designed in this paper. In particular, it can be seen from the actuator dynamics trajectories of the $x$-axis (Figure 6) that the method proposed in this paper has an obvious effect on weakening the system chattering. Compared with [30,31], the chattering amplitude is respectively reduced by more than 50%. Even after the fault occurs, the method in this paper has an obvious effect on fault compensation and weakens the local chattering caused by the fault.

From Figure 7 to Figure 8, we can see that, even after $k = 200$, the performance of the method in this paper is still better than that in [30,31], which is manifested in that the change of control law is relatively stable and convergence speed is significantly faster. In

Figure 7, we can see that the method in this paper converges and tends to be stable at around $k = 350$, while the method in Refs. [30,31] converges and tends to be stable when $k = 425$ and $k = 700$, respectively. The method in this paper can better compensate for the fault, weaken the chattering of the control law, and realize the fault-tolerant control more quickly.

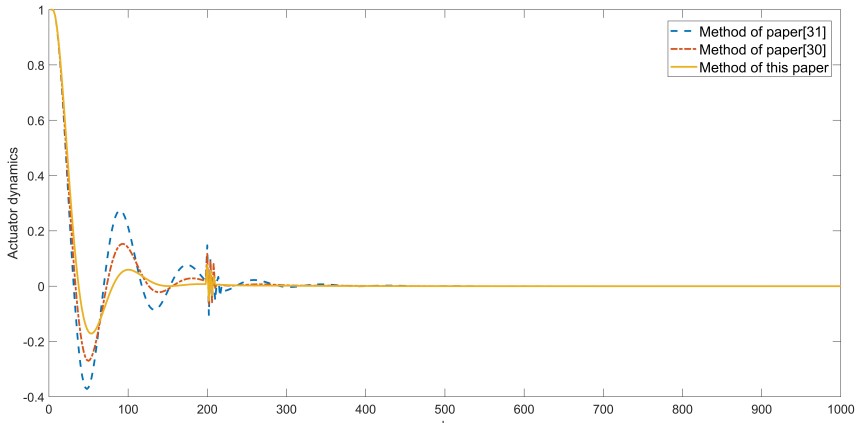

**Figure 6.** the actuator dynamics' trajectories.

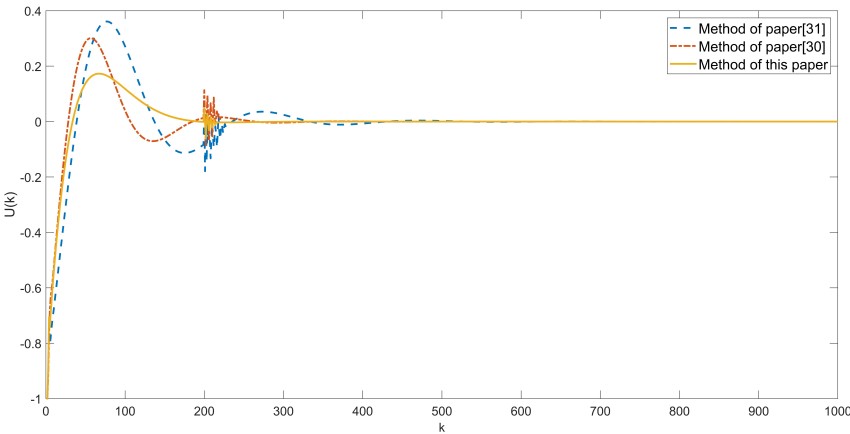

**Figure 7.** the trajectories of control law.

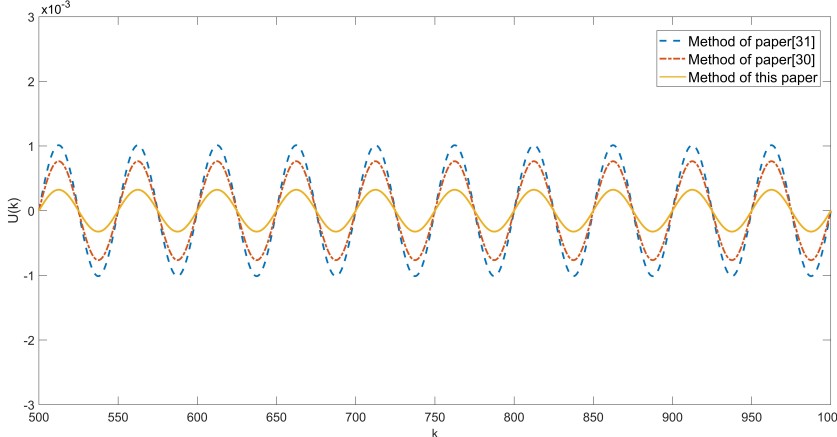

**Figure 8.** enlarged view of control law trajectories.

Figure 8 is an enlarged view of the control law. It can be seen that the proposed method significantly reduces the amplitude of chattering and makes the change of the control law

more gentle. Especially from Table 3, it can be concluded that the maximum amplitude of the method in this paper is weakened by about 68.04% and 57.7% compared with [30,31], respectively.

**Table 3.** Control law buffeting amplitude comparison.

| Method | Maximum Amplitude ($\times 10^{-3}$) |
|--------|--------------------------------------|
| Ref. [31] | 1.0124 |
| Ref. [30] | 0.7649 |
| This paper | 0.3235 |

To further illustrate the advantages of the method designed in this paper in dealing with input delay and state delay, we separately set up a set of the *x*-axis position curve comparison tests for time delays. The above simulation results are obtained when the state delay is set to $\tau_1 = 3$ and the input delay is $\tau_2 = 3$. In the case of other simulation conditions being the same, set $\tau_1 = \tau_2 = 0.6$ and $\tau_1 = \tau_2 = 5$, respectively; then, we obtain the simulation results in Figures 9 and 10. Compared with Figure 4, we can conclude that, when the time delay is small, all three methods can stabilize the flight state of the quad-rotor UAV in a short time; when the time delay is larger, the advantages of the method in this paper are apparent.

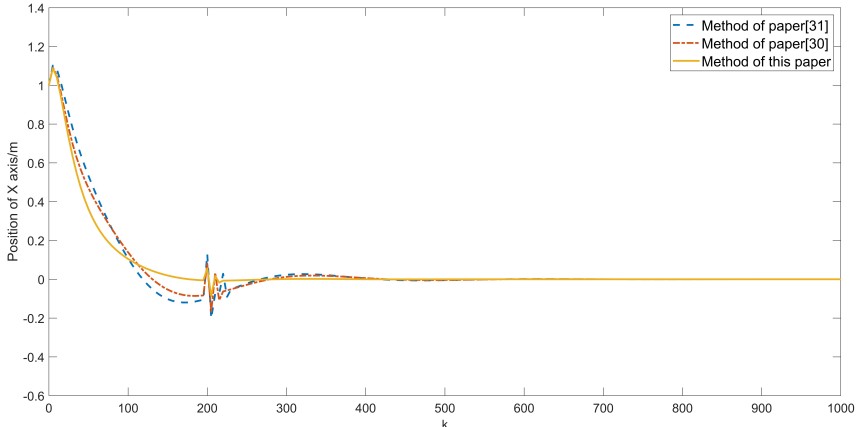

**Figure 9.** the position trajectories of the *x*-axis (2).

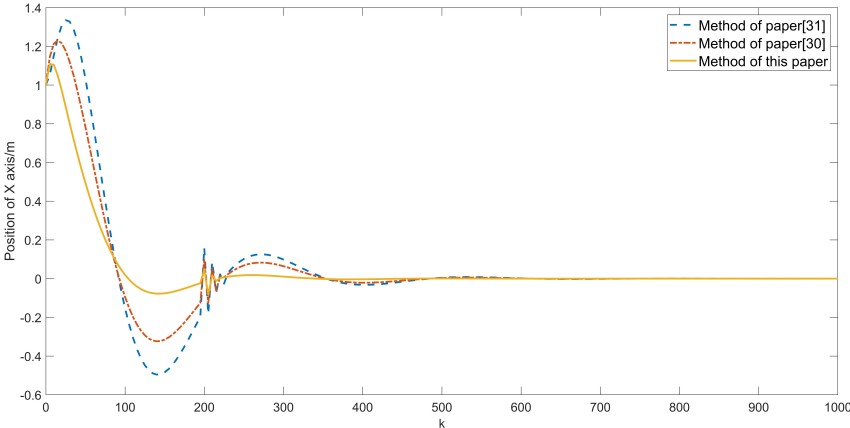

**Figure 10.** the position trajectories of the *x*-axis (3).

## 6. Conclusions

An intelligent SMP-FTC method has been investigated in this paper for uncertain discrete systems with sensor fault and external disturbances. In the design of the SMP

controller, a quasi-integral sliding mode surface has been used to design the SMP model, which ensures global robustness. Then, a double-power function with a novel compensation term has been designed as a reference trajectory, which effectively compensates the fault and time delays. An improved DIECOA of the rolling optimization is designed to guarantee the rapid convergence of the control law. Simulation results on the fault-tolerant simulation platform of the quad-rotor UAV show the effectiveness of the proposed method.

In this paper, only sensor faults have been considered; simultaneously, actuator and sensor faults will be considered in future works. Another direction worthy of research in the future is to take more challenging trajectories, such as ascending spiral, infinity, and complete physical verification.

**Author Contributions:** Conceptualization, P.Y.; methodology, Z.Z.; formal analysis, Z.Z.; investigation, Z.Z. and H.G.; writing—original draft, Z.Z.; writing—review and editing, Z.Z.; project administration, P.Y. and B.J.; resources, H.G. and X.H. All authors have read and agreed to the published version of the manuscript.

**Funding:** This research was funded by Key Laboratories for National Defense Science and Technology (6142605200402), the National Key Laboratory of Science and Technology on Helicopter Transmission (Grant No. HTL-O-21G11), the Aeronautical Science Foundation of China (20200007018001), the Aero Engine Corporation of China Industry-university-research cooperation project (HFZL2020CXY011), and the Research Fund of State Key Laboratory of Mechanics and Control of Mechanical Structures (Nanjing University of Aeronautics and Astronautics) (MCMS-I-0121G03).

**Institutional Review Board Statement:** Not applicable.

**Informed Consent Statement:** Not applicable.

**Data Availability Statement:** Not applicable.

**Conflicts of Interest:** The authors declare no conflict of interest.

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
