# Peer review of "Intelligent Discrete Sliding Mode Predictive Fault-Tolerant Control Method for Multi-Delay Quad-Rotor UAV System Based on DIECOA"

_aerospace, doi:10.3390/aerospace9040207_

Round 1
Reviewer 1 Report
In the paper: an intelligent SMP-FTC method has been investigated in this paper for uncertain discrete systems with sensor faults and external disturbances – based on a multicopter.
The authors need to be more careful to explain all abbreviations, when they first appear in the document e.g. UAV, FTC, TS T-S – also the writing of e.g. T-S should be consistent and the meaning should be explained
Chapter 2: where does this model come from? Was it already developed in literature? – this needs to be clarified.
The main weaknesses are in section 5. In line 194 as sensor fault function is given, but what does it mean?
- Which sensor is meant?
- Which signals is this sensor sending?
- Due to which physical or digital fault phenomenon could this sensor fault function be resulting?
These things need to be investigated and explained in detail – because without a realistic fault scenario the whole chapter is pointless.
Additionally, you talk about simulation experiments, but only simulation data is presented. You have the multicopter available, so please generate also experimental data and differentiate clearly.
In line 254 you write “Finally, on the fault-tolerant simulation platform of the quad-rotor UAV, the verification of the method in this paper is completed through comparative experiments.” – so these experiments (flying with the quadrocopter” need to be performed and evaluated – at least partly for checking the plausibility of the simulations.
Small remark: please check the document for missing blanks
Reviewer 2 Report
This paper aims to present a sliding mode predictive fault-tolerant control method based on the dynamic information exchange coyote optimization algorithm. This methodology is applied to a quadrotor UAV with multiple delays and sensor faults. In general, the paper has an adequate structure, however, the writing must be improved.
I suggest defining basic concepts of fault diagnosis and fault-tolerant control, e.g., a review of convex approaches for control, observation and safety of linear parameter varying and Takagi-Sugeno systems, processes.
Define acronyms and variables in first use, e.g. VOF. Check de manuscript thoroughly.
Most of the arguments about the contribution in this paper are based in that this paper includes time delays and it considers sensor faults instead of actuator faults which are more common in UAV papers. Include in the introduction the motivation for working with time delays for the UAV.
On the other hand, it is true that most of the papers about fault diagnosis and fault-tolerant control in UAVs consider actuator faults. However, there are a lot of approaches in other context that deals with sensor faults. Please mention the most relevant for your work.
Be clear in what papers are you referring to, there are some mentions to "the above papers" which is not clear.
I do not see what sensor faults and which types of faults are you considering in your problem. Please include the physical meaning.
In order to enrich the literature review you may discuss the following papers: Actuator fault detection and isolation on a quadrotor unmanned aerial vehicle modelled as a linear parameter-varying system, measurement and control; An integral TSMC-based adaptive fault-tolerant control for a quadrotor with external disturbances and parametric uncertainties, aerospace sci and tech; Actuator and sensor fault estimation based on a proportional multiple‐integral sliding mode observer for linear parameter varying systems with inexact scheduling parameters, ijrnc; Sensor fault-tolerant control of a quadrotor unmanned aerial vehicle, proc. inst meas eng E.
In the second section, the dimension of matrix A is wrong. The \Delta matrices are not perturbations, are uncertainty matrices.
Eq 3 is not a simplification of eq 2. You are rewriting the equation in a different form.
Include a detailed block diagram of the overall closed-loop system.
Include a flow diagram of the control algorithm.
In the results section include a more practical interpretation of the simulation scenario, faults and plots. How do they occur and at what time? what is the physical meaning of the faults?
How did you determine the uncertainty matrices?
Use more challenging trajectories, e.g. ascending spiral, infinity etc.
Use 3d plots in the result section.
Improve the conclusions, it reads like a summary. Include future proposed works.
The language must be improved, there are some typos, grammar mistakes that should be corrected. There are also some misused terms, e.g. "buffeting peak", "internal perturbation", etc.
Round 2
Reviewer 1 Report
Thanks to the sensible changes the paper can now be accepted.
Reviewer 2 Report
The paper has improved. However, in the light of the new information, I still have comments:
* I would include in the FTC definition: Possibly at a reduced level of performance depending on the severity of the fault, e.g. a review of convex approaches for control, observation and safety of linear parameter varying and Takagi-Sugeno systems, processes.
* Include a comment in the values chosen for the parametric uncertainty.
* What is randsin^2(k) for the disturbance (page 15, line 209)?
* The simulation must include both scenarios: the system operating fault-free and faulty. Therefore, the fault must be injected time after the simulation has started, in order to evaluate the complete FTC operation.
* The English still needs to be polished, e.g. "rewritted" should be "rewritten". Check the manuscript thoroughly.
Round 3
Reviewer 2 Report
I have no further comments